# Many-body density and coherence of trapped cold bosons

Camille Lévêque[1,2], Fritz Diorico[3], Jörg Schmiedmayer[1] Axel U. J. Lode[4*]

**1** Vienna Center for Quantum Science and Technology, Atominstitut, TU Wien,
Stadionallee 2, 1020 Vienna, Austria
**2** Wolfgang Pauli Institute c/o Faculty of Mathematics, University of Vienna,
Oskar-Morgenstern Platz 1, 1090 Vienna, Austria
**3** Institute of Science and Technology (IST Austria), Am Campus 1, 3400 Klosterneuburg,
Austria
**4** Institute of Physics, Albert-Ludwig University of Freiburg, Hermann-Herder-Strasse 3,
79104 Freiburg, Germany
* camille.leveque@sorbonne-universite.fr

February 23, 2021

## Abstract

**Many-body densities and correlation functions are of paramount importance for understanding quantum many-body physics. Here, we present a method to compute them; our approach is general and based on the action of bosonic or fermionic annihilation field operators on the many-body wavefunction. We analyze $N = 6$ quasi-one-dimensional harmonically-trapped bosons with weak to strong contact interaction strength up to the Tonks-Girardeau limit with infinite repulsion using the MultiConfigurational Time-Dependent Hartree method for indistinguishable particles (MCTDH-X). We compare our MCTDH-X solutions to the analytical ones in the infinite repulsion regime as well as to the so-called correlated pair wavefunction approach and find a good agreement. Since numerical approximations are not bound to the cases where analytical solutions are known, we thus demonstrate a general method to investigate high-order reduced density matrices and correlation functions in systems for which analytical solution are unknown. We trace the build-up of correlation features in the crossover from weak interactions to the Tonks-Girardeau limit and find that the higher-order correlation functions and densities resemble those in the Tonks-Girardeau limit for way smaller interactions than anticipated from just the one-body density.**

# 1 Introduction

Understanding a quantum many-body state implies to understand the role of correlations. The knowledge of the correlation functions at all orders is equivalent to solving the many-body problem [1, 2]. Correlation functions and their factorization quantify (high-order) coherence and the degree to which the (many-body) densities are representable as products of single-particle densities [3–5]. The degree to which one can extract high-order correlations and their information content [6–9] defines our knowledge about physical models of the many-body system.

Correlations have attracted considerable interest in various fields of physics from early-universe cosmology [10] via high-energy physics [11, 12] to imaging [13]. The experimental access to many-body correlation functions of correlated quantum matter [6,7,9,14–20] heralds the need for a general theoretical framework to evaluate them.

The Tonks-Girardeau (TG) gas is one of the handful of many-body systems for which an analytical solution is known. Using the Bose-Fermi mapping [21, 22] a system of bosons interacting via a contact potential in one-dimension can be mapped onto a non-interacting set of spinless fermions. This model was widely investigated theoretically and experimentally approached in the literature [23–29], but so far high-order many-body densities and correlations were not reported.

Here, we investigate the fermionization process of a few bosons in a one-dimensional (1D) harmonic trap by means of the high-order many-body densities and correlations functions. To this end we put forward, verify, and apply a general approach to numerically obtain these quantities for general many-body wavefunctions, following Ref. [30], that requires solely the application of the annihilation field operator to the given many-body state. The results presented here were obtained using the multiconfigurational time-dependent Hartree method for indistinguishable particles (MCTDH-X) [31–33] and we show that our results are in good

agreement with analytical predictions for finite interaction strengths as well as with numerical predictions by the correlated pair wavefunction method [34–36].

## 2 Methods and system

### 2.1 Many-body wavefunctions

Typical wavefunction-based numerical methods represent the wavefunction with number states (configurations) that are weighted with complex-valued coefficients, i.e.,

$$|\Psi(t)\rangle = \sum_{\vec{n}} C_{\vec{n}}(t)|\vec{n}\rangle, \tag{1}$$

providing a multiconfigurational ansatz built from a set of time-dependent coefficients $\{C_{\vec{n}}(t)\}$ and configurations $|\vec{n}\rangle$. These configurations are constructed from a set of (time-dependent) orbitals $\{\phi_i\}_{i=1}^M$ (in the MCTDH-X method [33, 37, 38]), such that

$$|\vec{n}\rangle = \mathcal{N} \prod_{i=1}^{M} \left[\hat{b}_i^\dagger\right]^{n_i} |vac\rangle. \tag{2}$$

Here, $\mathcal{N} = \left(\prod_{i=1}^M n_i!\right)^{-\frac{1}{2}}$ a normalization constant and $\hat{b}_j^\dagger$ ($\hat{b}_j$) the operator that creates (annihilates) a particle in the orbital $\phi_j(\chi_i)$. In the following we use $\chi_i$ as a general coordinate and specify $\chi_i = \mathbf{x}_i$ for position and spin and $\chi_i = \mathbf{k}_i$ for momentum and spin where applicable.

### 2.2 High-order densities and correlations

The creation [annihilation] field operator $\hat{\Psi}^\dagger(\chi_i)$ [$\hat{\Psi}(\chi_i)$] can be expanded in the basis set of orbitals,

$$\hat{\Psi}^\dagger(\chi_i) = \sum_{j=1}^{M} \hat{b}_j^\dagger \phi_j^*(\chi_i), \qquad \hat{\Psi}(\chi_i) = \sum_{j=1}^{M} \hat{b}_j \phi_j(\chi_i), \tag{3}$$

respectively. The diagonal of $p$-body density ($p$-BD) can be represented as the expectation value of a product of creation and annihilation field operators,

$$\rho^{(p)}(\chi_1, \ldots, \chi_p) = \langle \Psi | \hat{\Psi}^\dagger(\chi_1) \ldots \hat{\Psi}^\dagger(\chi_p) \hat{\Psi}(\chi_p) \ldots \hat{\Psi}(\chi_1) | \Psi \rangle. \tag{4}$$

This $p$-BD quantifies $p$-body correlation and coherence. The diagonal elements of the $p$-BD give the probability to find the particles $1, \ldots, p$ at $\chi_1, \ldots, \chi_p$. The effect of the interactions on the coherence of the many-body state can be quantified through the diagonal of the $p$th-order Glauber correlation function ($p$-GC),

$$g^{(p)}(\chi_1, \ldots, \chi_p) = \frac{\rho^{(p)}(\chi_1, \ldots, \chi_p)}{\prod_{i=1}^{p} |\rho^{(1)}(\chi_i)|}. \tag{5}$$

When the $p$-BD is a product of one-body density then $g^{(p)} = 1$ holds and the system is fully coherent; partial coherence ($g^{(p)} \neq 1$) is obtained when the $p$-BD is not a product of one-body densities. The $p$-GCs thus provide a spatially-resolved measure of coherence. For the sake

of notational convenience, we will omit the arguments in the $p$-BD and $p$-GC that we keep constant to obtain two-dimensional or one-dimensional visualizations [cf. labels $g^{(p)}(x_1, x_2)$ and $\rho^{(2)}(x_1, x_2)$ in Figs. 1, 2, 3]. For instance, in Figs. 1 and 3 we use $\rho^{(p)}(x_1, x_2)$ also for the cases where $p > 2$; we imply the cut $\rho^{(p)}(x_1, x_2, x_3 = x_3^{ref}, x_4 = x_4^{ref}, ...)$ of the $p$-BD $\rho^{(p)}$ [analogously for Figs. 2 and 3 with $g^{(p)}(x_1, x_2)$]. We provide the respective reference values $x_k^{ref}$ in the Figure's caption.

Both, $p$-BDs and $p$-GCs, can be evaluated from the wavefunction and the action of the annihilation field operators, Eq. (3), on it. The reduced wavefunctions of $N - k$ particles,

$$|\Psi^{(k)}\rangle = \begin{cases} |\Psi\rangle, \text{ if } k = 0 \\ \mathcal{N}_k \hat{\Psi}(\chi_k)|\Psi^{(k-1)}\rangle, \text{ if } k = 1, \dots, N - 1 \end{cases} \tag{6}$$

can be evaluated by *iteratively* applying the annihilation field operator to evaluate *successively* the coefficients of the $N - k$ bosons configurations until $k = p$. The inner product of the reduced $N - p$ bosons wavefunction provides the $p$-BD for a set of coordinates $\{\chi_i\}_{i=1}^p$. If different sets of coordinates are used for the creation and annihilation operators the off-diagonal elements of the $p$-BDs and $p$-GCs are obtained. A simpler way to evaluate the diagonal elements of $p$-BD uses the conditional densities $\rho_{\text{cond}}^{(j)}(\chi_j) = \langle\Psi^{(j)}|\hat{\Psi}^\dagger(\chi_k)\hat{\Psi}(\chi_k)|\Psi^{(j)}\rangle$ by noticing [30]

$$\rho_{num}^{(p)}(\chi_1, \dots, \chi_p) = \prod_{j=1}^p \rho_{\text{cond}}^{(j-1)}(\chi_j). \tag{7}$$

Here, $\rho_{\text{cond}}^{(0)}(\chi) = \rho^{(1)}(\chi, \chi)$ was used [cf. Eqs. (4) and (6)]. This provides an efficient and iterative way to evaluate the $p$-BD from a set of conditional lower-order densities; it is *not* a factorization of the $p$-BD in terms of lower-order ones – generally $\rho_{\text{cond}}^{(j)}(\chi) \neq \rho^{(j)}(\chi) \ \forall j > 0$. Note that the above equations hold for general indistinguishable particles, i.e., bosons and fermions.

## 2.3 System

We now consider the Tonks-Girardeau (TG) limit [21], for which the real-space wavefunction and energy of $N$ bosons become identical to the absolute value of the real-space wavefunction and energy of $N$ non-interacting fermions, respectively. Considering $N$ bosons in an harmonic trap in one dimension interacting via a contact potential of strength $\lambda$, the Hamiltonian reads,

$$\hat{H} = \frac{1}{2} \sum_{i=1}^N \left[ -\frac{\partial^2}{\partial x_i^2} + x_i^2 \right] + \lambda \sum_{i<j}^N \delta(x_i - x_j). \tag{8}$$

This system has the advantage that for $\lambda \to \infty$ the Bose-Fermi mapping provides an analytically exact solution for the wavefunction,

$$\begin{aligned} \Psi_{TG}(x_1, \dots, x_N) &= \prod_{1 \leq j < l \leq N} sgn(x_j - x_l)\Psi^F(x_1, \dots, x_N) \\ &= |\Psi^F(x_1, \dots, x_N)|. \end{aligned} \tag{9}$$

Here, $\Psi^F(x_1, \dots, x_N) = \frac{1}{\sqrt{N!}} \det[\phi_0(x_1) \dots \phi_{N-1}(x_N)]$ is the wavefunction of $N$ non-interacting fermions and $sgn(x)$ is the sign function. For a harmonic trap of frequency $\omega = 1$, the orbitals

$\{\phi_n(x)\}_{n=0}^{N-1}$, in the Slater determinant, read $\phi_n(x) = (2^n n! \sqrt{\pi})^{-\frac{1}{2}} H_n(x) e^{-\frac{x^2}{2}}$ and $H_n(x)$ are the Hermite polynomials. In the following, we consider $N = 6$ bosons and we compare the analytical $p$-BD, $\rho_{TG}^{(p)}$, obtained from integrating $|\Psi_{TG}(x_1, \ldots, x_N)|^2$ in $N - p$ coordinates, and the ones obtained using the MCTDH-X method via Eq. (7).

## 3  Results

### 3.1  Higher-order densities in the TG limit

We now apply our approach and compute the $p$-BDs in real space ($\chi = x$) from the MCTDH-X wavefunction (see also appendix A). Fig. 1 depicts a comparison of the results obtained for $\rho_{num}^{(p)}$ and $\rho_{TG}^{(p)}$ for $2 \leq p \leq 6$. The $p$-BD provides the probability of detecting particles at the position $x_1$ and $x_2$, given that the remaining $p - 2$ particles are fixed at some chosen reference positions (see caption of Fig. 1). For fermionized bosons, the diagonal of the high-order $p$-BD $\rho^{(p)}(x, x)$ vanishes. This so-called *correlation hole* results from the infinite (or large for numerical results) value of the interaction strength that mimics the Pauli principle, preventing to find two bosons at the same position. Moreover, $\rho^{(p)}(x_1, x_2)$ remains localized at the center of the trap, $|x_i| \lesssim 3.5$, because of the finite energy of the system $E_{TG} = \frac{N^2}{2}$ for $\lambda \to \infty$. The maxima are well defined in a peaked structure, indicating the localization of the atoms in position space. The maxima are either along the anti-diagonal ($x_1 = -x_2$), the bosons maximize the distance between each other, or along the sub-diagonal, to minimize the potential energy. For $p > 2$, correlation holes additionally appear at the fixed values of the remaining $p - 2$ coordinates of $\rho^{(p)}$, preventing to find other bosons at these positions.

The numerical and analytical results are in very good agreement, concerning both the amplitude and the features of $\rho^{(p)}$ for all orders, with an error, $|\Delta\rho^{(p)}|^2 = |\rho_{num}^{(p)}(x_1, x_2) - \rho_{TG}^{(p)}(x_1, x_2)|^2$, that remains below $2 \times 10^{-2}$. An analysis of the differences between $\rho_{num}^{(p)}$ and $\rho_{TG}^{(p)}$ shows that $\rho_{num}^{(p)}$ maxima and minima appear at slightly shifted values of $x_1$ and $x_2$ compared to the analytical ones. These shifts (also observed in the density, see appendix B) yield the largest values of $|\Delta\rho^{(p)}|^2$. Generally, the underestimation of the correlation in the MCTDH-X wavefunction predicts atom's positions further apart in comparison to the analytical wavefunction. Despite that the TG limit is particularly challenging for the MCTDH-X ansatz Eq. (1), quantitative information can be extracted from the numerical evaluation of the $p$-BDs from Eq. (7), which relies solely on the accuracy of the wavefunction.

### 3.2  Higher-order Glauber correlations in the TG limit

The $p$-GC are evaluated using Eq. (5) and provide spatially-resolved information about the coherence of the system. A comparison of $g^{(p)}(x_1, x_2)$ between analytical (lower triangle) and numerical (upper triangle) results for $2 \leq p \leq 6$ is provided in Fig. 2. The MCTDH-X wavefunction overestimates coherence, but – not surprisingly – the correlation holes remain in the $p$-GC. The two- and three-body coherence remains ($g^{(p)} \sim 1$ for $p = 2, 3$), while the higher-order many-body coherence diminishes as the order is increased for the cuts presented ($g^{(p)} < 0.6$ for $p = 4$ and $g^{(p)} < 0.4$ for $p = 5, 6$). The coherence observed in low-order $p$-GC can be explained by the population of the first natural orbitals that scales as $\sim \sqrt{N}$ [39]. The discussion about the differences between the analytical and numerical results on $p$-BD remain

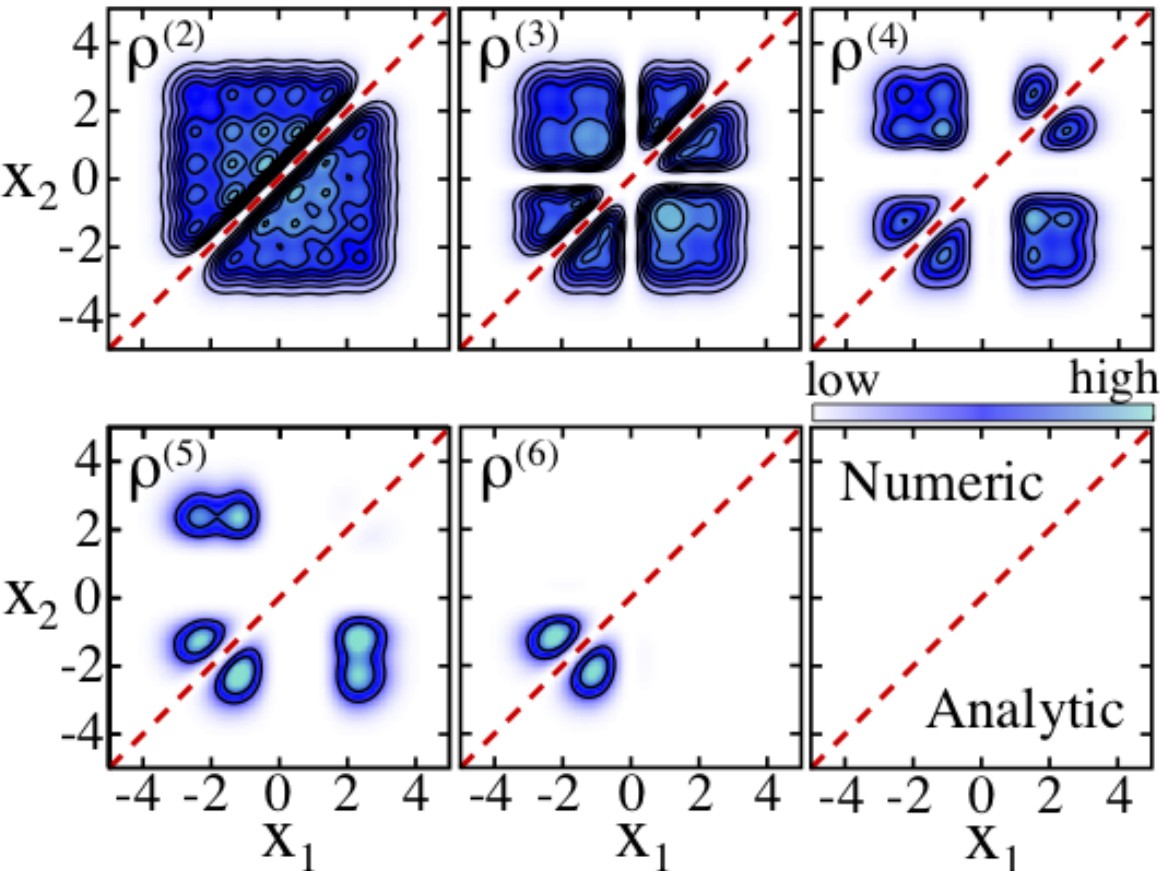

Figure 1: Many-body densities $\rho^{(p)}$, with $p = 2, \ldots, 6$ for $N = 6$ bosons in the Tonk-Girardeau limit $\lambda \to \infty$. The $p$-BDs are symmetric with respect to the diagonal, $x_1 = x_2$ (dashed red line). The numerical results are obtained with MCTDH-X for $M = 24$ orbitals and a contact interaction strength $\lambda = 1500$ and are depicted on the upper diagonal ($x_2 > x_1$). The analytical results, obtained using the Bose-Fermi mapping [see Eq. (9) and text below] are on the lower diagonal ($x_1 > x_2$). For $p > 2$, the $p$-BDs are plotted fixing all coordinates except two, i.e. $\rho^{(3)}(x_1, x_2, 0)$, $\rho^{(4)}(x_1, x_2, 0, 0.47)$, $\rho^{(5)}(x_1, x_2, 0, 0.47, 1.03)$ and $\rho^{(6)}(x_1, x_2, 0, 1.03, 1.5, 2.53)$. The thin black isolines on the plot of $\rho^{(p)}$ are equally spaced by 0.15 for $p \leq 4$ and 0.1 for $p > 5$. On the error plot, the isolines are separated by 0.005, except for $p = 6$ where 0.0005 is used.

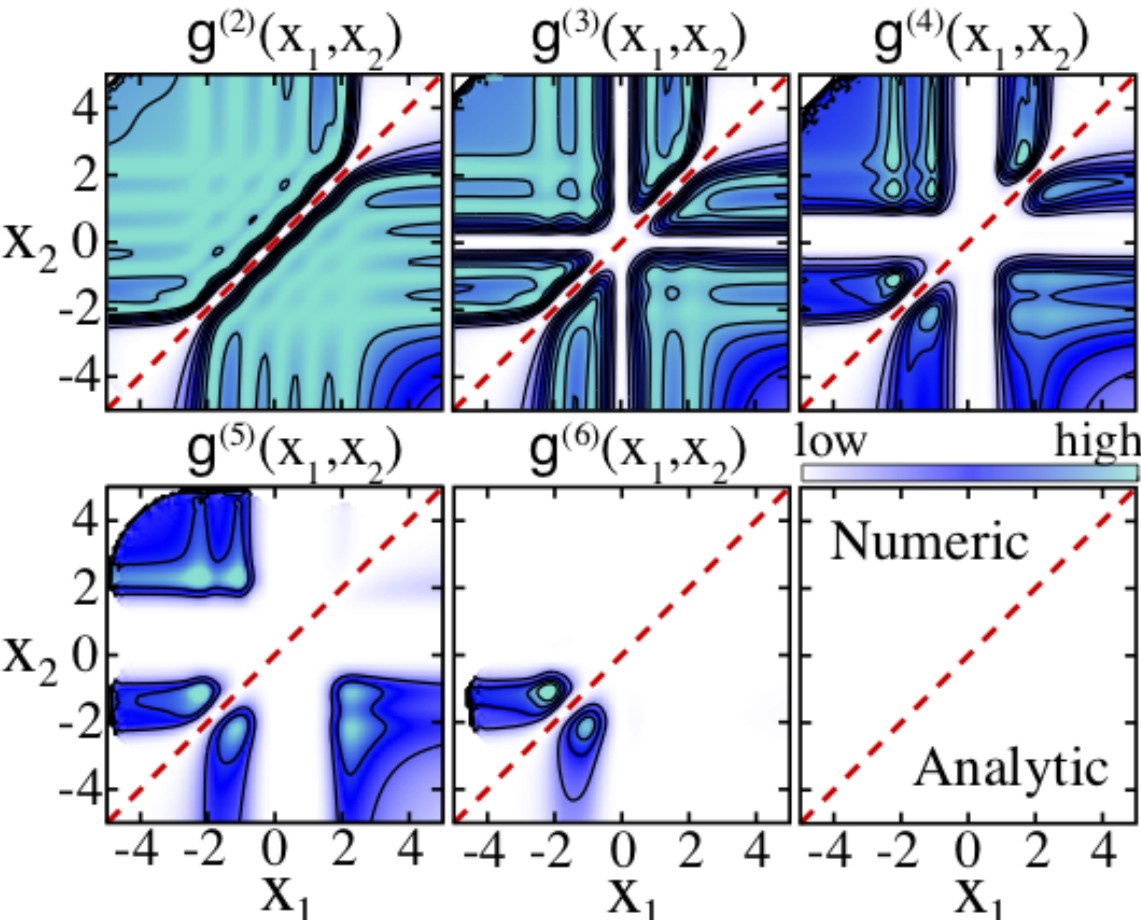

Figure 2: Glauber correlation functions $g^{(p)}$ [Eq. (5)] for $N = 6$ bosons in the Tonks-Girardeau limit. The $p$-GC are symmetric with respect to the diagonal, $x_1 = x_2$ (dashed red line), thus numerical results, obtained with MCTDH-X using the same parameters than in Fig. 1, are depicted on the upper diagonal $(x_2 > x_1)$, the analytical results are on the lower diagonal $(x_1 > x_2)$. For $p > 2$. We fix some coordinates in the $p$-GC analogous to the $p$-BD, see caption of Fig. 1.

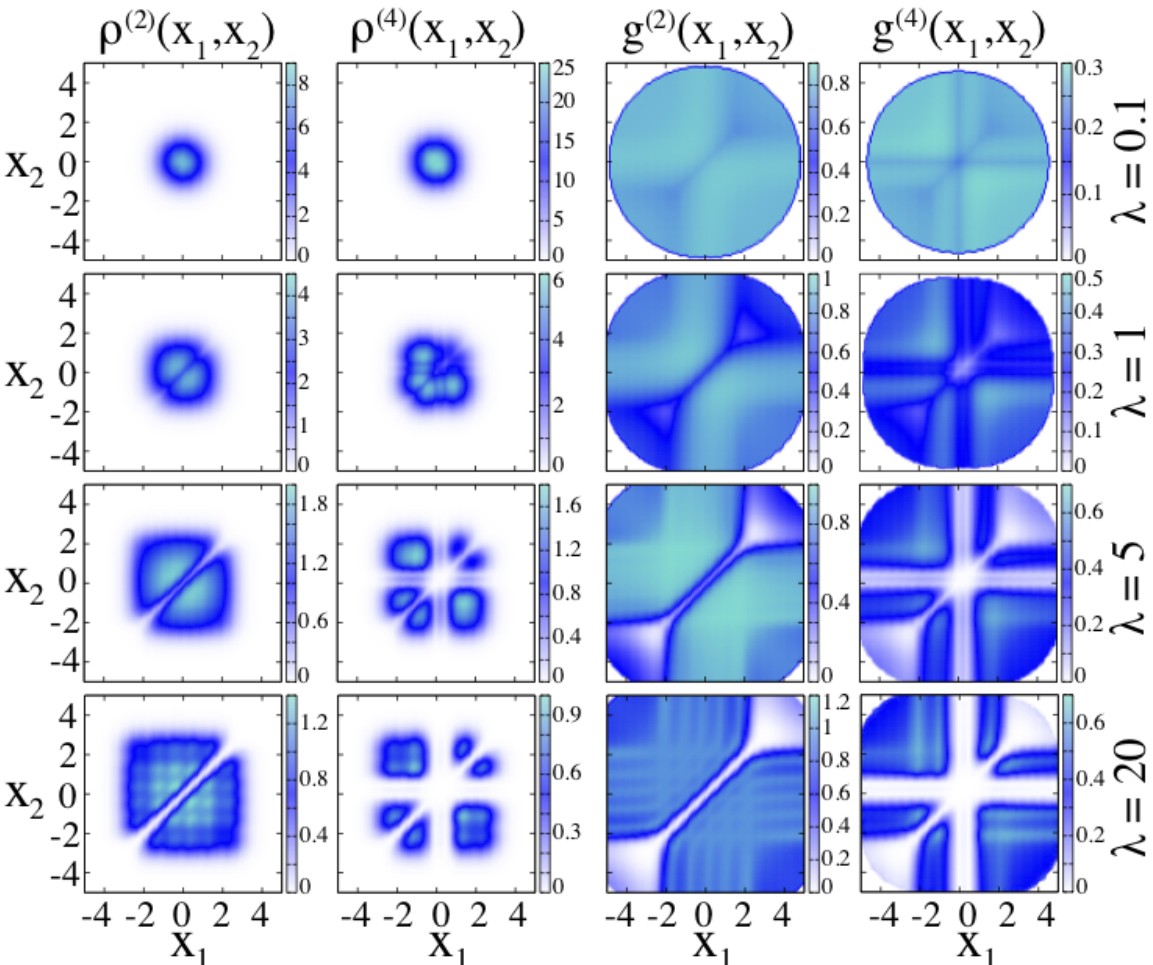

Figure 3: Two-dimensional cuts of the $p$-BD $\rho^{(p)}(x_1, x_2)$ (first two columns) and $p$-GC $g^{(p)}(x_1, x_2)$ (last two columns), with $p = 2$ and 4 for $N = 6$ bosons for different interaction strength $\lambda$ from 0.1 to 20 (rows). The results are obtained using MCTDH-X with $M = 22$ orbitals for $0.1 \leq \lambda \leq 5$ and $M = 24$ for $\lambda \geq 20$. For order $p > 2$, the $p$-BD and $p$-GC are plotted fixing all coordinates except two. For $\lambda \leq 0.1$ the coordinates are all fixed at $x_i^{ref} = 0$, for $\lambda \geq 1$ the coordinates are fixed to $x_3^{ref} = 0$, $x_4^{ref} = 0.47$. To avoid numerical errors, the plot of the $p$-GC was restricted to coordinates $(x_1, x_2)$ where the denominator of Eq. (5) was larger than $10^{-10}$.

true for the $p$-GC. Note that similar numerical errors in the position of maxima and minima appear for the 1-BD as well as the $p$-BD, the numerical $p$-GC reproduce the features of the exact $p$-GC particularly well.

 

# 4 Results: comparison of MCTDH-X and correlated pair wavefunctions

In the Tonks-Girardeau limit, analytical results are available, and thus numerical methods are not strictly necessary in this regime. The purpose of numerical methods is to investigate many-body systems when analytical results are unknown. It is important to assess the accuracy of numerical approximations where no analytical solutions are available. To this end, we confirm the accuracy of the MCTDH-X approach in a benchmark using the correlated pair wavefunction method [35, 36], among others [40], and for a wide range of interactions.

In this correlated pair wavefunction approach, the many-body wavefunction is expanded using parabolic cylinder functions $D_\mu$,

$$\Psi_{CP}(x_1, \ldots, x_N) = C \prod_{j<l}^{P} D_\mu(\beta|x_j - x_l|)e^{-\frac{NR^2}{2}}, \tag{10}$$

Here $P = \frac{N(N-1)}{2}$ is the number of distinct pairs, $R = \frac{1}{N}\sum_{i=1}^{N} x_i$ is the center of mass, $C$ is a normalization constant and the two parameters $\beta = \sqrt{\frac{2}{N}}$ and $\mu$ solve the transcendental equation,

$$\frac{\lambda}{\beta} = -\frac{2^{3/2}\Gamma(\frac{1-\mu}{2})}{\Gamma(\frac{-\mu}{2})},$$

fixed by the boundary condition at $|x_j - x_l| = 0$. This wavefunction is exact for $\lambda = 0$ and $\lambda \to \infty$ and was shown to provide consistent results compared to numerical methods for few particles for the energy and the density of the ground state.

The high-order density matrices, $\rho_{CP}^{(p)}$, are obtained from the numerical integration of the CP wavefunction over $N - p$ coordinates. To easily compare with the results obtained with MCTDH-X, we introduce the relative difference,

$$\int_{-L}^{L} |\rho_{CP}^{(p)}(x_1, \ldots, x_p) - \rho_{num}^{(p)}(x_1, \ldots, x_p)|dx_1 dx_2. \tag{11}$$

Here, $L$ defines the grid used for the computations and $p - 2$ coordinates are fixed at reference values. The results obtained for different interaction strengths from 0.01 to 100 are reported in Fig. 4.

For weak interactions, $\lambda \le 0.1$, the agreement between the MCTDH-X and CP results is very good for the fourth-, fifth- and sixth-order density depicted in Fig. 4.

The discrepancy increases for larger values of $1 \le \lambda \le 5$. It reaches a maximum and decreases again for $\lambda > 5$. This comparison shows that for weakly interacting bosons, the MCTDH-X method provides accurate results as the CP approximation was shown to be accurate for the weakly interacting regime. Moreover, the different numbers of orbitals ($12 \le M \le 22$) do not affect the results much; thus, the calculations are very close to converged. The relative difference between the TG prediction and the CP approach lies below 0.05 for $\lambda = 100$; accurate results are expected for the CP approach in this regime. The relative differences obtained from the comparison of MCTDH-X with $M = 22$ and the CP approach lie around 0.15, slightly larger than for $\lambda = 0.1$. The different number of orbitals provide different relative differences, but the accuracy for $M = 22$ very good.

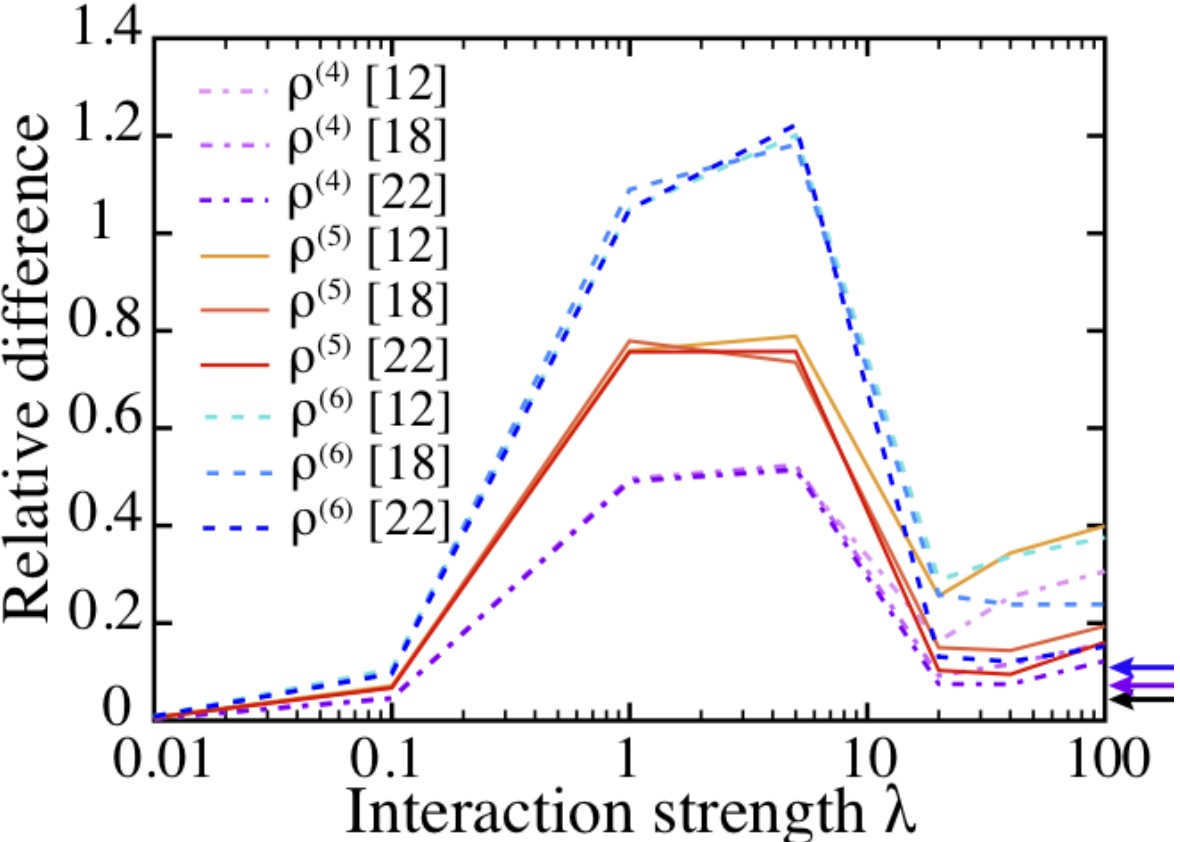

Figure 4: Relative difference between $\rho_{CP}^{(p)}$, obtained using the CP wavefunction [see Eq. (10)], and $\rho_{num}^{(p)}$ obtained using MCTDH-X [see Eq. (7) in the main text] using $M = 12, 18,$ and 22 orbitals, indicated in square brackets. The figure shows the relative difference [see Eq. (11)] for the orders $p = 4$, dotted-dashed line [purple], $p = 5$, full line [orange] and $p = 6$ dashed line [blue] as a function of the interaction strength $0.01 \leq \lambda \leq 100$. The black arrow indicates the relative difference between $\rho_{CP}^{(p)}$ and $\rho_{TG}^{(p)}$ for $\lambda = 100$ and $4 \leq p \leq 6$, that lies on the thickness of the arrow. The purple arrow indicates the relative difference between $\rho_{num}^{(p)}[M = 24]$ and $\rho_{TG}^{(p)}$ for $\lambda = 100$ and $p = 4$, and the blue arrow for $p = 5$ and 6. The reference positions are chosen as $x_3 = 0$, $x_4 = 0.46875$, $x_5 = 1.03125$ and $x_6 = 1.5$.

The most interesting results of this comparison are obtained for the intermediate inter-action strength $1 \leq \lambda \leq 5$. For these interactions the values of the relative difference are the highest for all orders $p$ of the density. Moreover, the number of orbitals used seems to be sufficient to converge the MCTDH-X calculations, because the results are only slightly different for all cases considered, $12 \leq M \leq 22$. Thus, these results indicate that the CP approach is less accurate than MCTDH-X for this range of interactions, explaining the large values for the relative difference. To support this conclusion, we plot a cut of the density $\rho^{(p)}$ for $4 \leq p \leq 6$ in Fig. 5. We can see that for small relative differences, i.e. $\lambda \leq 0.1$ and $\lambda \geq 20$, the MCTDH-X results converge to the CP results by increasing the number of orbitals. But for $1 \leq \lambda \leq 5$, the MCTDH-X results do not converge to the CP density, but to other values. Thus, by virtue of the variational principle, these results show that the MCTDH-X gives more accurate results than the CP approach for this regime of interactions.

## 4.1 High-order densities and Glauber correlations in the crossover from weak to strong interactions

Assured by the accuracy of our approach, we now investigate the build-up of correlations in the crossover from weak to strong interactions. In Fig. 3 we report the many-body densities and correlations, respectively, for $\lambda = 0.1, 1, 5$ and 20. For weak interactions ($\lambda = 0.1$) the $p$-BDs remain Gaussian-like for all orders, see appendix C for complementary results for $\rho^{(3)}, \rho^{(5)}, \rho^{(6)}$. The effects of the interactions are only grasped in the $p$-GC that exhibit a loss of coherence, even for $\lambda = 0.1$, when more atoms are fixed at $x^{ref} = 0$, indicating that the atoms' positions depend on each other. For $\lambda = 1$, $\rho^{(2)}(x_1, x_2)$ starts to exhibit a square shape at the center of the trap and the diagonal starts to be depleted. When $\lambda$ is increased to 5, the correlation hole on the diagonal is formed and $\rho^{(2)}(x_1, x_2)$ reaches out to coordinates where the potential has larger values, while for $\lambda = 20$ the peak structure appears similar to the TG results. For higher-order, $g^{(p)}$ with $p > 2$, correlation holes become more pronounced as the $p$-BDs gradually converge to the TG limit.

The second-order $p$-GC reveal that the coherence is maintained ($g^{(2)} \sim 1$) across the whole system for $\lambda \leq 5$, with $g^{(2)}(x_1, x_2) \sim 1$. For $\lambda = 20$ bunching ($g^{(2)}(x_1, x_2) > 1$) emerges for specific $x_1 \neq x_2$ with a spatial structure that is similar to the TG limit (compare 2 and 3), indicating the localization of the atoms. For higher orders, $g^{(p)}(x_1, x_2)$, with $p > 2$ we see that for $\lambda = 1$ correlations holes are not yet formed properly, but the spatial structure of correlations is similar to the TG limit. The difference with the TG limit becomes much smaller for $\lambda = 5$ and almost vanishes for $\lambda = 20$. These findings are reminiscent to what was known for the 1-body and 2-body densities of few-body systems [41–43], and our results show that it remains true for higher-order of the $p$-GC and $p$-BD. Thus high-order $p$-GC and $p$-BD can be used to assess the when the strong interaction limit is reached and the transition from few to many body systems.

For strong interactions, our results for the $p$-BD and $p$-GC resemble the expected results from the TG regime for strong interactions, while the dimensionless total energies are still rather different from the TG limit – 13.2, 17.13 and 18 for $\lambda = 5, 20$ and $\infty$, respectively. For $\lambda = 5$, the features of $p$-BD and $p$-GC are still distinct from the TG limit, but for $\lambda = 20$ it may be rather difficult to experimentally distinguish between finite interactions and the TG limit. Inherent experimental inaccuracies arise from the shot noise of finite-particle-number-systems and from the finite number of single-shot images available to measure high-order $p$-BDs and $p$-GCs. Nonetheless, spatially resolved $p$-BD and $p$-GC should be possibly obtained

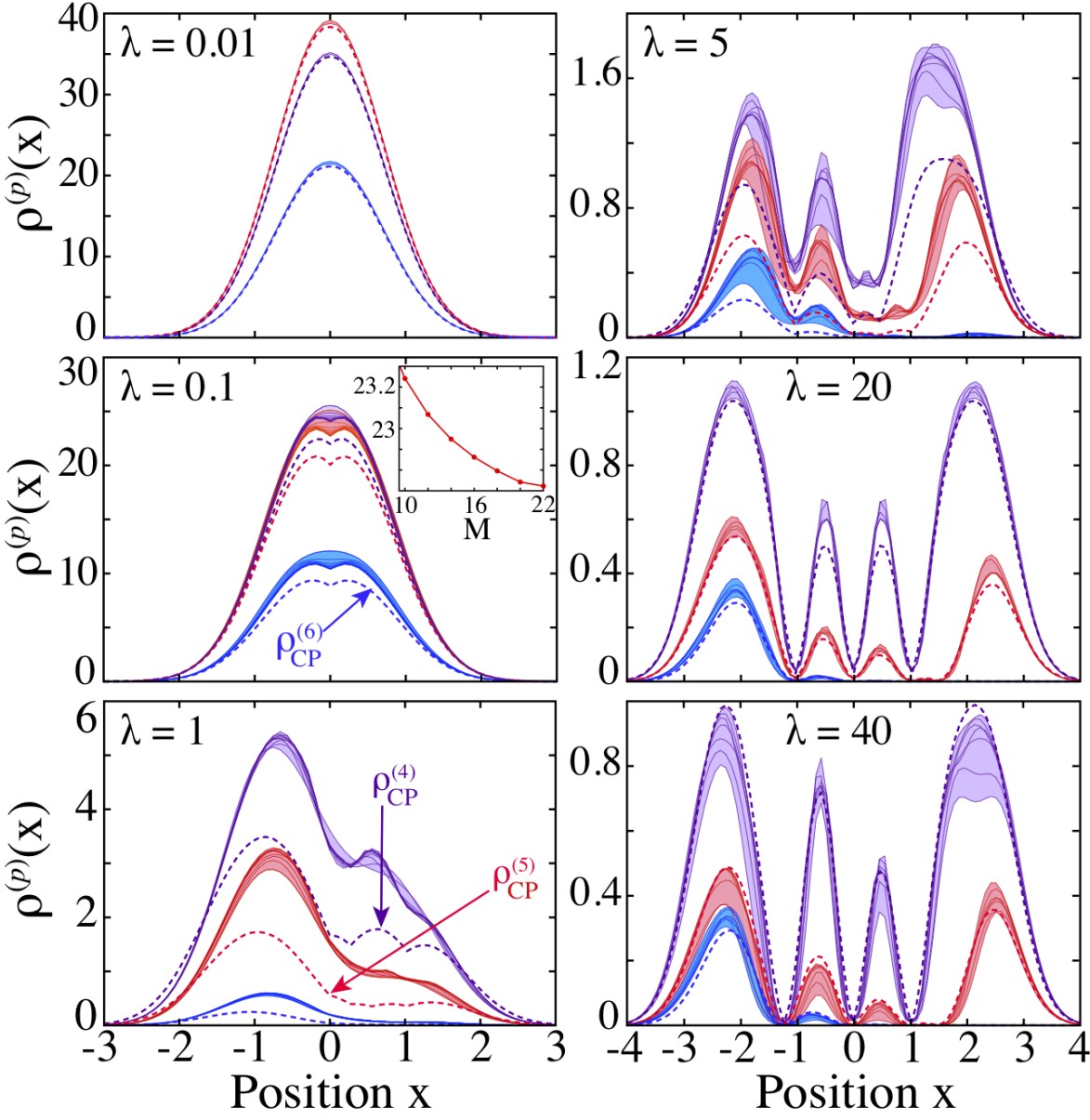

Figure 5: One-dimensional cut of the high-order density $\rho_{CP}^{(p)}$ using the CP wavefunction, dashed line, for $p = 4$ in purple, $p = 5$ in red and $p = 6$ in blue. The shaded (colored) areas indicate the value of $\rho_{num}^{(p)}$ obtained with MCTDH-X for different numbers of orbitals thin full lines (see text for the values of $M$), same color code as $\rho_{CP}^{(p)}$. The inset in $\lambda = 0.1$ shows the value of $\rho_{num}^{(5)}$ at the cusp in $x = 0$ as a function of the number of orbitals.

experimentally for Tonks gas [44, 45].

## 5   Conclusions

We introduced a general method to compute the $p$-BD and $p$-GC of *any order* for *general* many-body states in second-quantized representation including, for instance, exact diagonalization of Bose-Hubbard models [46, 47], the density matrix renormalization group methods reviewed in Refs. [48, 49], or other methods in the MCTDH-X family [50–53]. As an application, we first benchmarked and then explored the $p$-BD and $p$-GC of a few bosons in the crossover from weak interactions to the Tonks-Girardeau limit of infinite repulsion; for our comparison, we used analytical results as well as the correlated pair wavefunction approach. We demonstrate that the higher-order correlation functions and densities resemble those in the Tonks-Girardeau limit for way smaller interactions than anticipated from just the one-body density.

Our present approach thus opens up a new way to explore many-body physics in great detail in an hitherto under-explored field that became only recently accessible experimentally [6–9]. Many applications are viable: the many-body correlations of vortices [54], bosons in double well potentials [55], tunneling to open space [56], with long-range interaction [57, 58], multi-component bosons [59] bosons immersed in high-finesse optical cavities [60], trapped fermions [61] or mixtures of particles [62–64].

## Acknowledgements

We thank Ofir E. Alon and João Sabino for useful comments and discussions in the elaboration of this work.

**Funding information**   This work was supported by The Wiener Wissenschafts- und Technologie Fonds (WWTF) project No. MA16-066. AUJL acknowledges financial support by the Austrian Science Foundation (FWF) under grant No. P-32033-N32. CL acknowledges funding by the FWF under grant No. M-2653. Computation time on the Hazel Hen and Hawk computers at the HLRS Stuttgart is gratefully acknowledged.

## A   Multiconfigurational time-dependent Hartree method for bosons

In the main text, we use the Multiconfigurational time-dependent Hartree method for bosons [33, 37, 38, 55, 56] (MCTDH-B) implemented in the MCTDH-X software [31, 32, 59, 61]. MCTDH-B is a wavefunction-based approximation to solve the time-dependent many-body Schrödinger equation. The MCTDH-B method relies a multiconfigurational ansatz to approximate the many-body wavefunction,

$$|\Psi(t)\rangle = \sum_{\vec{n}}^{N_{conf}} C_{\vec{n}}(t)|\vec{n};t\rangle, \tag{12}$$

where $C_{\vec{n}}(t)$ are the time-dependent expansion coefficients in the basis of the time-dependent configurations $|\vec{n};t\rangle$. For bosons, each configuration is a permanent, i.e., a fully symmetric arrangement of $N$ particles in a set of $M$ time-dependent orbitals $\{\phi_i(x,t)\}$. In turn, the orbitals are expressed in a time-independent or *primitive* basis. The equations of motion for the time-dependent orbitals and coefficients are obtained via the time-dependent variational principle [65]. Both, the coefficients and the orbitals are optimized at all points in time to minimize the error with respect to the exact wavefunction. In principle, for a sufficiently large number of orbitals the MCTDH-B wavefunction, Eq. (12), converges to the exact wavefunction [51,53,61,66]. The number of orbitals needed to converge a calculation depends on the physical problem considered.

MCTDH-B was introduced to solve time-dependent problems but using Wick's rotation $t \rightarrow -i\tau$, i.e., a propagation with imaginary time, the ground state of the Hamiltonian considered can be obtained using an initial guess. In the main text, we consider the ground state of $N = 6$ bosons in the Tonks-Girardeau limit, i.e., we approach the limit of bosons with infinite repulsion, cf. Figs. 1 and 2 in the main text. The results in the main text were obtained with $M = 24$ orbitals expressed in a plane wave (FFT) primitive basis with 256 basis functions and a spatial domain of $[-12, 11.9]$.

## B    Density in position and momentum space

Here we compare the density, $\rho(x) = \langle \Psi | \hat{\Psi}^\dagger(x) \hat{\Psi}(x) | \Psi \rangle$, obtained from MCTDH-B calculations to analytical results. As in the main text, we consider $N = 6$ bosons in the Tonks-Girardeau limit, i.e., with infinitely strong repulsive contact interactions. The density in position space can be obtained analytically (see main text) in this limit and can thus be used to assess the accuracy of numerical calculations. In Fig. 6, we can see that increasing the number of orbitals in MCTDH-B computations makes the density closer to the exact one. Due to the non-analytical sign function [cf. main text Eq. (9)] the results remain substantially different even for $M = 24$ orbitals.

Nonetheless, the salient features of the density are obtained for all numbers of orbitals depicted, i.e., there are as many peaks as particles and the density is centered at the minimum of the harmonic trap at $x = 0$. Interestingly, the MCTDH-B density minimizes the energy by increasing the spacing between the bosons (peaks) as compared to the exact density. This suggests that the number of orbitals remains too small to capture the correlation between the bosons entirely.

In momentum space, the density, $\rho(k) = \langle \Psi | \hat{\Psi}^\dagger(k) \hat{\Psi}(k) | \Psi \rangle$, is known to exhibit a cusp at $k = 0$ [34,67]. This cusp is specific to bosons in the Tonks-Girardeau limit; the momentum density of non-interacting fermions is generally different from the momentum density of bosons in the Tonks limit.

## C    Complementary results for the crossover from weak to strong interactions

Here, we provide additional results for the crossover between weak to strong interaction presented in the main text, in which orders $p = 2$ and $4$ of the many-body density ($p$-BD)

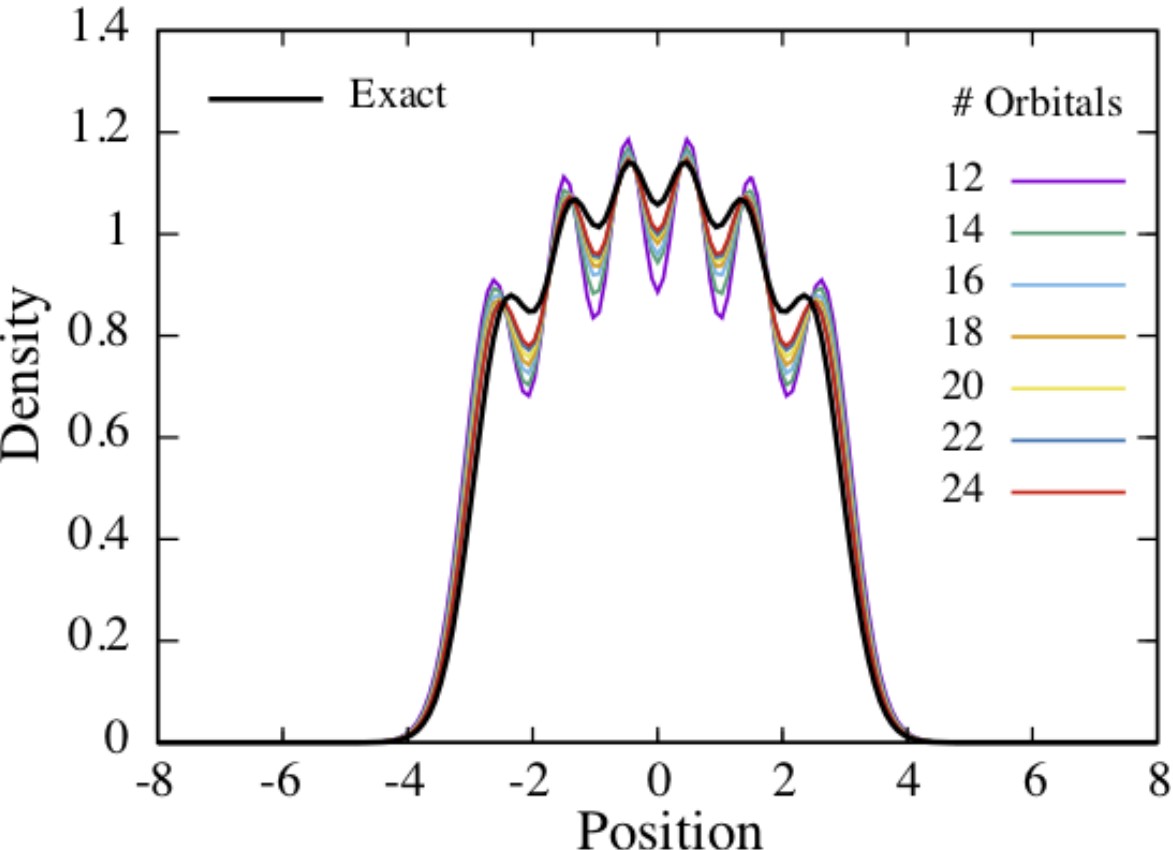

Figure 6: Density $\rho(x)$ for $N = 6$ bosons in the Tonk-Girardeau limit $\lambda \to \infty$. The analytical result (thick black line) is compared to numerical results obtained with MCTDH-X for $M = 12$ to 24 orbitals, with $\lambda = 1500$. By increasing the number of orbitals, the density resembles the analytical results more and more, but does not completely converge to it for M=24. More orbitals are needed at this strong interaction for a higher degree of convergence.

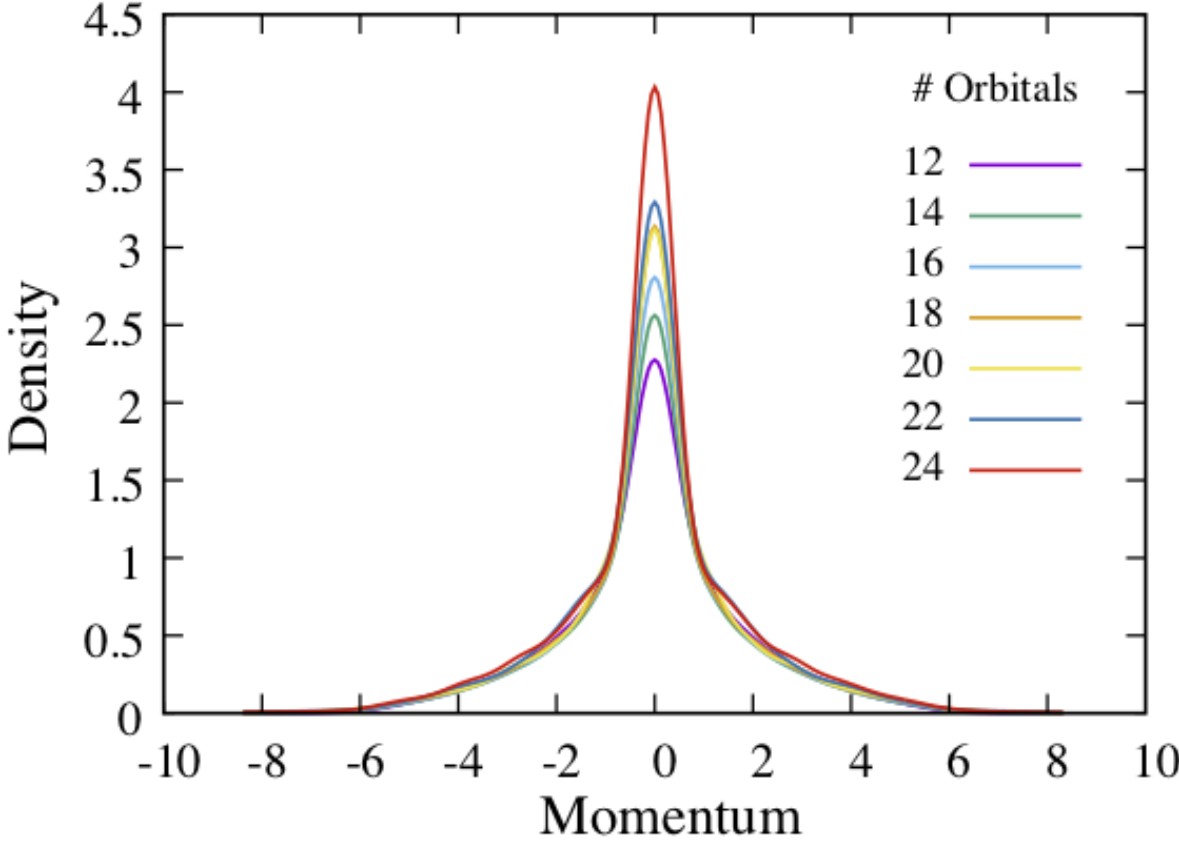

Figure 7: Density $\rho(k)$ for $N = 6$ bosons in the Tonk-Girardeau limit $\lambda \to \infty$. The numerical results are obtained with MCTDH-X for $M = 12$ to $24$ orbitals, with $\lambda = 1500$. For an increasing number of orbitals, the cusp at $k = 0$ in the density appears.

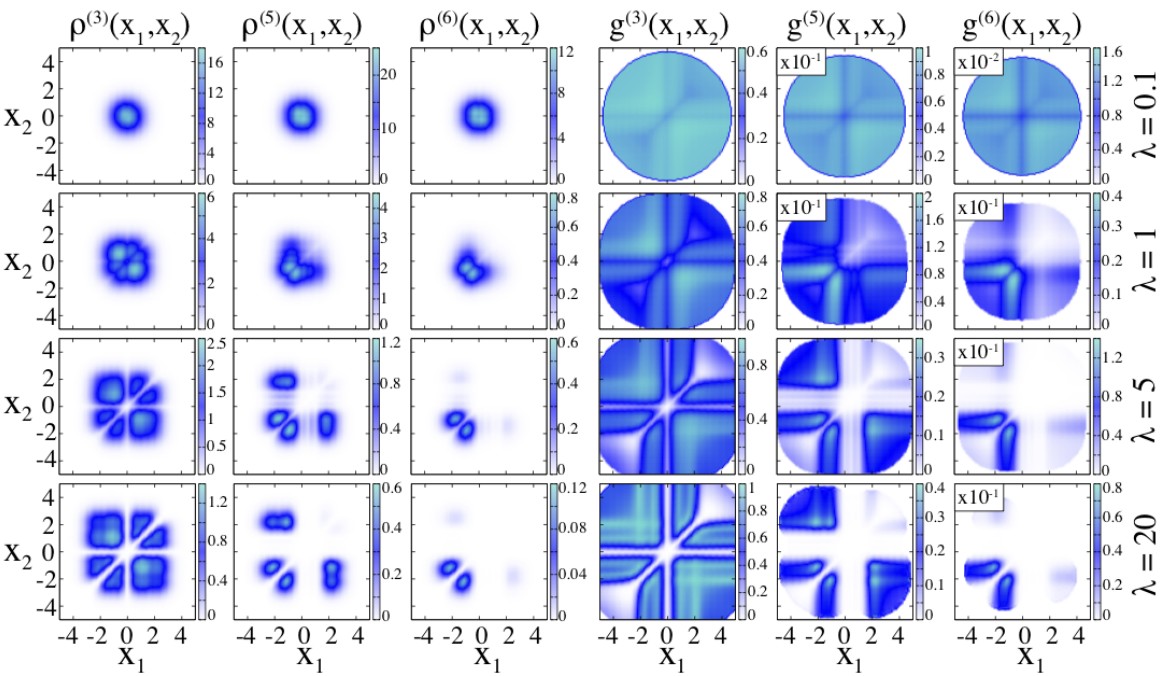

Figure 8: Two-dimensional cuts of the $p$-BD, $\rho^{(p)}(x_1, x_2)$, (first three columns) and $p$-GC, $g^{(p)}(x_1, x_2)$, (last three columns) with $p = 3, 5$ and 6 for $N = 6$ bosons for different interaction strength $\lambda$ from 0.1 to 20 (rows). The results are obtained using MCTDH-X with $M = 22$ orbitals for $0.1 \leq \lambda \leq 5$ and $M = 24$ for $\lambda \geq 20$. The $p$-BD and $p$-GC are plotted fixing all coordinates except two. For $\lambda \leq 0.1$ the coordinates are all fixed at $x_i^{ref} = 0$, for $\lambda \geq 1$ the coordinates are fixed to $x_3^{ref} = 0$, $x_4^{ref} = 0.47$, $x_5^{ref} = 1.0$ and $x_6^{ref} = 1.5$. To avoid numerical errors, the plot of the $p$-GC was restricted to coordinates $(x_1, x_2)$ where the denominator of Eq. (5) (main text) was larger than $10^{-10}$.

$\rho^{(p)}(x_1, x_2)$ and Glauber correlation ($p$-GC) $g^{(p)}(x_1, x_2)$ are presented. In Fig. 8, we report the $p$-BD and the $p$-GC for $p = 3, 5$ and 6, for the interaction strength $\lambda = 0.1, 1, 5$ and 20. For weak interaction ($\lambda = 0.1$) the $p$-BDs remain Gaussian-like for all orders. For $p \geq 5$ the effects of the interactions appear in the $p$-BDs and detecting additional atoms at $x^{ref} = 0$ becomes less likely. This effect is better grasped in the $p$-GC that exhibit a loss of coherence when more atoms are fixed at $x^{ref} = 0$. For $\lambda = 1$, the diagonal starts to be depleted at all orders $p$, as observed in the $p$-BDs and the $p$-GCs. The $p$-BDs remain clearly different to the Tonks gas limit while the features of $p$-GCs are rather similar despite their smaller values. When $\lambda \geq 5$, the correlation holes in the $p$-BD and the $p$-GC are formed at all orders and the results become increasingly similar to the TG limit.

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
