# Peer review of "Many-body density and coherence of trapped cold bosons"

_SciPost Physics_

## Round 2 · Referee Report · Anonymous · 2021-4-22

Strengths

- tackles a hard problem: high order correlations in a many-body quantum system

Weaknesses

- quality of presentation is low: the new method is not explained clearly enough

- the numerical checks are performed only for N=6 particles, making it impossible for the reader to appreciate the relevance and applicability of the method to true quantum many-body problems (i.e. with large N)

- it is unclear whether the new method, which relies on formula (7), would apply to more general correlation functions (e.g. off-diagonal $p$-body density matrix)

Report

After a short introductory discussion of higher-order correlations in many-body quantum systems, and a brief discussion of second quantization, the authors present their main formula (7), which is a recursive formula for the $p$-point correlation function of the operator $\Psi^\dagger(\chi) \Psi(\chi)$ (where $\chi$ is an arbitrary coordinate, for instance position or momentum).

Then, as far as I understand, formula (7) is exploited in conjunction with an approximate numerical method dubbed 'MCTDH_X', introduced elsewhere. Not much is explained about the 'MCTDH_X' method in this paper. Numerical results obtained from that method for the harmonically trapped delta Bose gas are compared to the exact solution in the limit of infinite repulsion for $N=6$ bosons, and also to another approximate method ('CP wavefunction') for weak repulsion.

In my opinion, the new method is not presented clearly enough. Formula (7), which is presented as the key formula of the paper, is a straightforward consequence of the definitions. What seems difficult is to evaluate the factors $\rho_{\rm cond}^{(j-1)}(\chi_j)$, yet the authors do not explain at all how this is done. The paper is not self-contained, and I find that it is simply not possible to understand how the 'new method' works, with the informations provided in the manuscript. I cannot recommend publication of this work in SciPost in the present form.

Requested changes

- if the main result of the paper is a new method for computing correlation functions, then the method should be presented in a clear and self-contained way

- the 'new method' seems to be very close to the one of Ref. [30], therefore the results of Ref. [30] should be recalled first, and then the differences or improvements brought by this new work should be explained clearly to the reader

- if the authors insist on presenting this method as a new tool for many-body systems, then it should be benchmarked against some other results for more than $N=6$ particles (one suggestion would be to use DMRG for the 1D delta Bose gas, which allows to do a few dozens of particles, see e.g. [Schmidt and Fleischhauer, Phys. Rev. A 75, 021601(R) 2007] or [Peotta and di Ventra, Phys. Rev. A 89, 013621, 2014])

- overall, the quality of the presentation is low and must be largely improved. In particular, in section 2 the authors should be more careful about their notations and the discussion of their formulas. For instance, they should say explicitly that $\mathcal{N}_k$ in Eq. (6) is fixed so that the wavefunction is normalized, and then they should give the relation between $\mathcal{N}_k$ and $\rho_{\rm cond}^{(j)}(\chi_j)$ (the indices are clearly wrong before Eq. (7) and must be fixed) and then they should unravel the mechanism to arrive at formula (7). Formula (7) must be properly discussed: why is it interesting and useful? In formula (7), what is $\rho_{\rm num}^{(p)} (\chi_1,\dots, \chi_p)$ exactly? Is it $\rho^{(p)} (\chi_1,\dots, \chi_p)$? If so, why changing notations? Sections 3 and 4 could also be improved and be made more reader-friendly. For instance, the new physics predictions or new effects investigated could be highlighted more, and could be presented in a section separated from the discussion of the accuracy of the method.

---

## Round 2 · Referee Report · Anonymous · 2021-6-10

Report

In the manuscript, the Authors provide a method to calculate many-body correlation functions for one-dimensional trapped bosons based on Multi-Configurational Time-Dependent Hartree method for indistinguishable particles (MCTDH-X). The method is demonstrated for both hardcore and softcore trapped bosons by calculating p-body density correlations (p-BD) as well as p-th order Glauber correlation functions (p-GC). By comparing the present results with exact analytical solutions (in the hardcore limit) and with the results from correlated pair wavefunction method (in case of softcore bosons), the Authors have provided validity as well as accuracy of their method.

Although the present method is another way to obtain many-body correlation functions, I cannot recommend the manuscript for publication in SciPost Physics, as the present results are immediate applications of MCTDH-X presented in Refs. [32, 33]. Moreover, the Authors have presented the results only for $N=6$ bosons, for which other methods, like DMRG, provide accurate results. It is also unclear the numerical complexity or computational cost of the present method for large $N$. To reiterate the Comment of Referee 1, it is hard for the reader to appreciate this MCTDH-X based method over other known algorithms (DMRG, Monte-Carlo etc.) in large $N$ setting. Therefore, I believe that the manuscript, in its present form, does not fulfill the acceptance criteria put forward by SciPost Physics.

Requested changes

There are a few shortcomings of the manuscript that may be improved.

1- The Authors should improve the presentation of Sec. 2. Specifically, the presentation of the paragraph containing Eqs. (6) and (7) should be improved. In this Section, the Authors should also describe the MCTDH-X method to make the manuscript self-contained.

2- Figs. 1 and 2 does not have any scale to the numerical data. Although the caption of Fig. 1 gives some details of the contour lines, but such presentation is difficult to grasp. Fig. 2 does not even contain such details about the contour lines. I suggest to provide proper numerical scale like in Fig. 3.

3- The caption of Fig. 1 tells “On the error plot, the isolines are separated by 0.005, except for p = 6 where 0.0005 is used”. However, there is no error plot in Fig. 1. In Sec. 3.1, the Authors claim that the error remains below $2 \times 10^{-2}$, but this is hard to understand from the plot.

4- The Authors should describe the computational cost of this method with increasing $N$ as well as with increasing number of orbitals used.

---

## Editorial Decision

awaiting_resubmission